# Early Detection of *Dendroctonus valens* Infestation at Tree Level with a Hyperspectral UAV Image

Bingtao Gao [1], Linfeng Yu [1,2], Lili Ren [1,3], Zhongyi Zhan [1] and Youqing Luo [1,3,*]

1 Beijing Key Laboratory for Forest Pest Control, Beijing Forestry University, Beijing 100083, China
2 School of Ecology and Nature Conservation, Beijing Forestry University, Beijing 100083, China
3 Sino-French Joint Laboratory for Invasive Forest Pests in Eurasia, Beijing Forestry University-French National Research Institute for Agriculture, Food and Environment (INRAE), Beijing 100083, China
* Correspondence: youqingluo@126.com; Tel.: +86-10-6233-6840

**Abstract:** The invasive pest *Dendroctonus valens* has spread to northeast China, causing serious economic and ecological losses. Early detection and disposal of infested trees is critical to prevent its outbreaks. This study aimed to evaluate the potential of an unmanned aerial vehicle (UAV)-based hyperspectral image for early detection of *D. valens* infestation at the individual tree level. We compared the spectral characteristics of *Pinus tabuliformis* in three states (healthy, infested and dead), and established classification models using three groups of features (reflectance, derivatives and spectral vegetation indices) and two algorithms (random forest and convolutional neural network). The spectral features of dead trees were clearly distinct from those of the other two classes, and all models identified them accurately. The spectral changes of infested trees occurred mainly in the visible region, but it was difficult to distinguish infested from healthy trees using random forest classification models based on reflectance and derivatives. The random forest model using spectral vegetation indices and the convolutional neural network model performed better, with an overall accuracy greater than 80% and a recall rate of infested trees reaching 70%. Our results demonstrated the great potential of hyperspectral imaging and deep learning for the early detection of *D. valens* infestation. The convolutional neural network proposed in this study can provide a reference for the automatic detection of early *D. valens* infestation using UAV-based multispectral or hyperspectral images in the future.

**Keywords:** red turpentine beetle (*Dendroctonus valens* LeConte); early detection; hyperspectral image; random forest; deep learning

## 1. Introduction

Forests play an essential role in terrestrial ecosystems, providing various ecological services such as water resource conservation, erosion control, climate change mitigation, and carbon sequestration. Forests face numerous stresses from biotic and abiotic factors, which are intensified by climate change [1]. On the one hand, more extreme weather events increase the vulnerability of trees. On the other hand, the suitable area and survival rate of pests have increased because of higher average temperatures [2]. The red turpentine beetle (RTB; *Dendroctonus valens* LeConte) is considered a secondary pest in its area of origin—North and Central America. However, it became a pine killer after invading China due to suitable climate and living conditions, causing huge economic and ecological losses [3]. RTB spread to north-eastern China in 2017. By 2018, its presence had been detected over an area of 156,000 hectares, infesting 338,000 pines in the region. Furthermore, the range and intensity of RTB outbreaks are expected to increase with global warming [4].

As an important invasive pest in China, RTB has become a primary pest, preferring to attack mature Chinese pine (*Pinus tabuliformis* Carr.) with a large diameter [5]. RTB attack usually occurs one generation per year in northeast China. Adults begin to emerge from

the roots in late spring and extensively attack the lower stem of new hosts [6]. Mid-to-late May and early–to-mid June is the peak of adult flight. Eggs are laid beneath the bark and hatch into larvae in about half a month. Larvae feed on the phloem and cambium of trunks and roots, forming common galleries. Larval feeding and associated fungal infection block the tree's conduction tissue [7]. The canopy color of infested trees changes gradually over time, from green to yellow, red and finally gray (i.e., needleless). Our field investigation found that the crown response is far hysteretic to the initial infestations, with noticeable fading usually occurring in the second year's growing season (Figure 1), but by this time most RTB have completed their development and spread to other trees. Therefore, the early identification and disposal of infested trees is essential to avoid the spread and outbreak of the infestation.

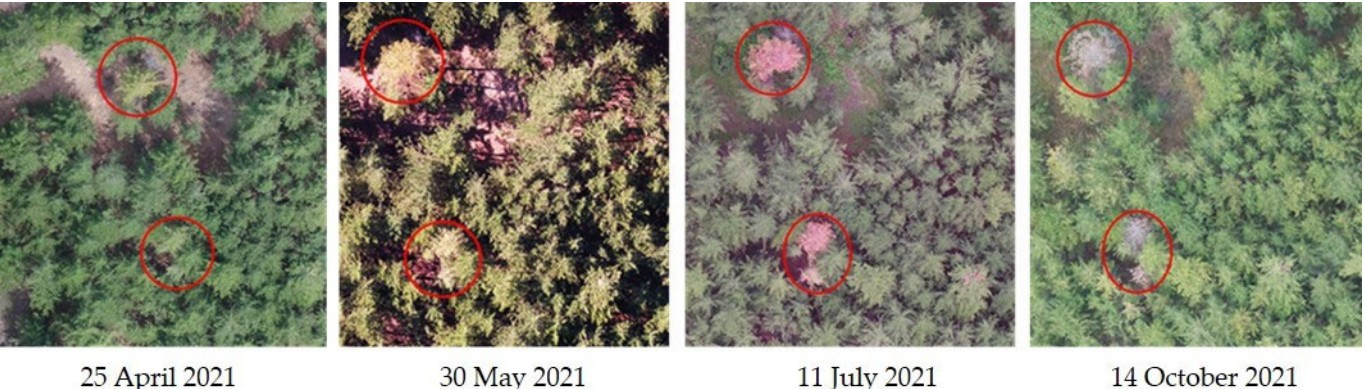

**Figure 1.** Crown color change of infested trees in UAV images. The red circles indicate pine trees infested by *D. valens*.

Remote sensing technology is an effective means of monitoring forest health and has been widely used. To date, multispectral satellite imagery has been successfully used to monitor the dynamics of bark beetle outbreaks and detect tree mortality induced by bark beetles [8–15]. However, it is still difficult to detect the early stage of bark beetle infestation in individual trees using satellite imagery due to spatial, temporal and spectral resolution restrictions [16]. Hyperspectral data can capture more detailed spectral information; therefore, their potential for early stress detection in vegetation has attracted attention. Several studies attempted to detect the early stages of bark beetle infestations using airborne hyperspectral data, but their accuracy was not sufficient for forestry practices, presumably due to the coarse spatial resolution [2,17–19]. Nasi et al. [20] compared the detection ability of unmanned aerial vehicle (UAV) hyperspectral datasets with ground sample distances (GSDs) of about 0.1 m and aircraft hyperspectral datasets with GSDs of 0.5 m for spruce bark beetle (SBB), which were obtained by a tunable Fabry–Pérot interferometer (FPI) camera with a wavelength range of 500–900 nm. The results showed that the finer resolution UAV hyperspectral datasets had higher classification accuracy (81%) when using a support vector machine (SVM) classifier to distinguish different infestation stages of SBB. However, the healthy class they defined contained no attack and green attack stages. Honkavaara et al. [21] utilized UAV-based multi-temporal hyperspectral images captured by a FPI camera with a spectral range of 502–907 nm and a GSD of 8.6 cm, and multispectral images with five bands and GSDs of 6.0 cm, to detect the early stage of SBB infestation, and found that hyperspectral data could improve the classification accuracy, although the overall accuracy was only 40–55%. Hellwing et al. [19] used indices based on high-spatial-resolution hyperspectral data (0.3 m) obtained using a hyperspectral HySpex camera with a wavelength range of 400–1000 nm to detect early SBB infestations, and obtained excellent results (overall accuracy = 98.84%) with the threshold method. Yu et al. [22] used the random forest (RF) algorithm and indices based on UAV hyperspectral imagery, with 281 channels from 400 to 1000 nm and a ground resolution of 0.4 m, to classify the

stages of pine wilt disease (PWD) infection of *P. tabuliformis*, and the overall accuracy was 74.38%. Einzmann et al. [23] studied the temporal changes in spectral features (reflectance, derivatives and vegetation indices) of Norway spruce after ring-barking and found that a vitality decline was detected earlier in hyperspectral data collected by two individual HySpex sensors (spectral ranges of 416–992 nm and 995–2498 nm with GSDs of 0.5 m and 1 m, respectively) than in field inspections. The classification accuracy of RF was 79% at the beginning of the second vegetation period, and the derivatives were the most important features. These studies indicate that high spatial resolution hyperspectral data have great potential for detecting early infestations of bark beetles at the individual tree level.

A hyperspectral image (HSI) is a three-dimensional dataset containing both spectral and geometric information. Research on HSI classification has received widespread attention in the field of remote sensing [24]. In the early stage of HSI classification research, many methods based on pixel-wise spectral signals were proposed, which could be divided into two strategies: spectral-matching-based and statistical-feature-based. Spectral-matching-based methods identify and distinguish different types of ground objects by matching the spectral curves of known and unknown targets, and commonly used methods include spectral angle mapper (SAM), spectral information divergence (SID), binary coding (BC), etc. [25]. However, these methods have poor discrimination ability for pixels with similar spectral curve shapes, and are easily affected by intra-class differences, which increases the difficulty in setting matching thresholds. The classification methods based on statistical features, such as maximum likelihood classification (MLC), artificial neural networks (ANNs), support vector machines (SVMs), decision trees (DCs), etc., form decision boundaries from training samples, and then use these decision boundaries for category prediction [25,26]. The variables used by these methods usually require prior expertise and mainly focus on spectral information while ignoring geometric information. Recently, deep learning has made great breakthroughs in the fields of big data analysis and computer vision, and has been introduced into HSI classification, including methods such as stacked auto-encoders (SAEs), convolutional neural networks (CNNs), deep belief networks (DBNs), recurrent neural networks (RNNs), generative adversarial networks (GANs), active learning models (ALs), etc. [24,27]. Among them, CNNs are widely used in the extraction of joint spatial–spectral information for HSI classification with superior performance [27–34]. In agricultural production, HSI analysis based on CNNs has been applied to variety classification [35,36], ripeness and component prediction [37] and pest and disease detection [38–40]. In forest applications, CNNs have also been successfully used for tree species classification with HSIs, yielding better results than traditional machine learning algorithms [41–45]. Compared with tree species classification, classifying infested trees at different stages is more challenging, especially at the early stage, when there are only subtle spectral differences. The CNN-based models proposed by Safonova et al. [46] and Nguyen et al. [47] were able to detect leaf discoloration or loss in fir trees caused by insects in high-spatial-resolution RGB images, but were powerless for early detection. Minařík et al. [48] compared the ability of three CNN architectures to detect individual trees infested by bark beetles in multispectral images, and all models could detect infested trees (F-score > 0.9). Yu et al. [49] developed a 3D-Resnet-CNN model to detect PWD-infected pine trees using a HSI and achieved an accuracy of 72.68% for early stage detection. However, the model was pixel-based (spatial resolution of 0.44 m), and the input was consisted in small neighboring regions (size = 11 × 11), which may not have been wide enough to resolve individual trees in high-spatial-resolution UAV images, with a spatial resolution of 0.01 m to 0.1 m [39].

To date, no study has used HSI to detect early infestation of RTB at the individual tree level. This study aims to evaluate the potential of UAV-based HSI to detect early RTB infestations. We first investigated the changes in the spectral traits of *P. tabuliformis* after a RTB infestation. Secondly, we compared the performance of different spectral features in distinguishing the stages of RTB using a RF classifier. Finally, we explored whether a deep

learning algorithm (CNN) outperformed a machine learning algorithm (RF) in classifying bark beetle disturbances in a small HSI dataset.

The main contributions of this study could be summarized as follows:

(1) We identified the changes in spectral features of *P. tabuliformis* infested by RTB, and established a classification model based on spectral vegetation indices (SVIs) and RF, which could provide a reference for monitoring damages of RTB using multispectral UAV or satellite data.

(2) A CNN architecture containing three types of Inception-resnet blocks was proposed to extract the joint spatial–spectral information from high-spatial-resolution HSIs and classify pine trees into different damage categories, which could be used for early detection of damage caused by RTB and other conifer-infesting wood-borer insects using UAV-based hyperspectral and multispectral images in the future.

## 2. Materials and Methods

### 2.1. Study Area and UAV-Based HSI Acquisition

The study site (approximately 118.97°E, 40.95°N) is located at the intersection of the three provinces of Liaoning, Hebei and Nei Mongol, and is affiliated with Lingyuan City, Liaoning Province, China (Figure 2). It is characterized by a mid-latitude temperate continental monsoon climate, with an annual average temperature of 8.7 °C and an annual average precipitation of 479.4 mm. The Chinese pine forests in Lingyuan City have suffered a massive RTB attack since 2017. The dominant tree species at the study site is the Chinese pine.

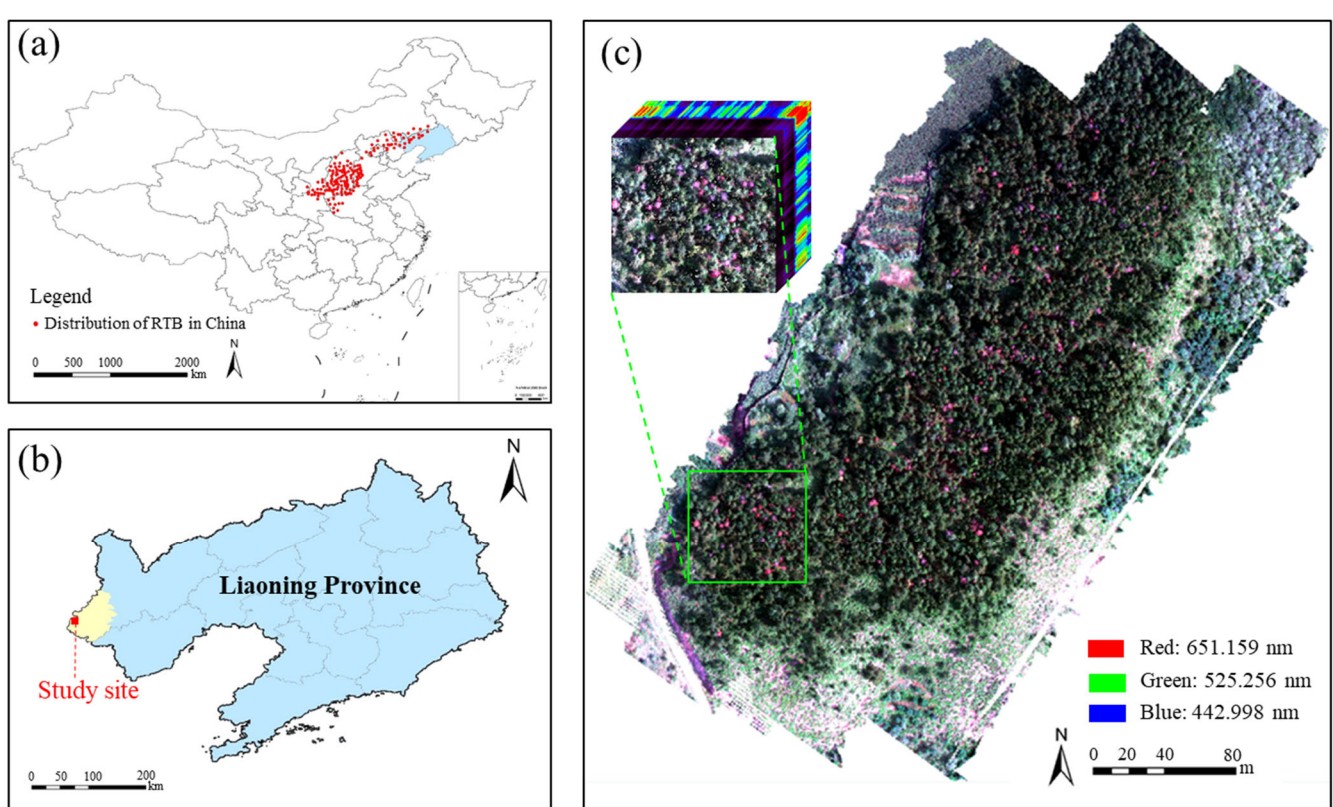

**Figure 2.** Study area. (**a**) The distribution of RTB and the location of Liaoning Province in China; (**b**) location of the study site; (**c**) UAV-based HSI of the study stand.

HSIs were acquired using a DJI Matrice 600 UAV (DJI, Shenzhen, China) equipped with a Pika L hyperspectral camera (Resonon, Bozeman, MT, USA). The flight mission was carried out from 10:00–11:00 on the morning of 23 August 2021. The flight altitude was 100 m, and the forward and side overlaps of HSI were 50%. The weather was cloudless

during the flight. A radiation-calibrated standard tarp was laid on the ground in the flight area. The HSIs included 150 spectral channels from 386 to 1025 nm with about 4 nm spectral resolution. The preprocessing main included radiometric calibration, reflectance correction, geometric corrections, hyperspectral orthophoto mosaics and spatial resolution resampling, as detailed in [49]. A Savitzky–Golay filter was performed in ENVI 5.3 with the default parameters to reduce noise. Finally, a hyperspectral image with 150 channels and 0.1 m spatial resolution was obtained, covering an area of about 7 hectares of forest (Figure 2c).

### 2.2. Ground Survey and Health Status Classification

The ground truth data were collected in early August 2021, two months after the peak period of adult RTB flight. A flight mission was carried out using the Inspire-2 drone with a Zenmuse X5S camera (DJI, Shenzhen, China) to obtain high spatial resolution (4 cm/pixel) orthophotos as reference data prior to ground surveys. Based on our previous survey results (Figure 3), the pine trees were divided into three groups: healthy trees, infested trees and dead trees. The indicators for assessing the health status of *P. tabuliformis* and the number of samples used in the next section are shown in Table 1 [50]. The ground survey results were annotated in RGB images captured by an Inspire-2 drone (Figure 4) for matching with the HSI in the next step.

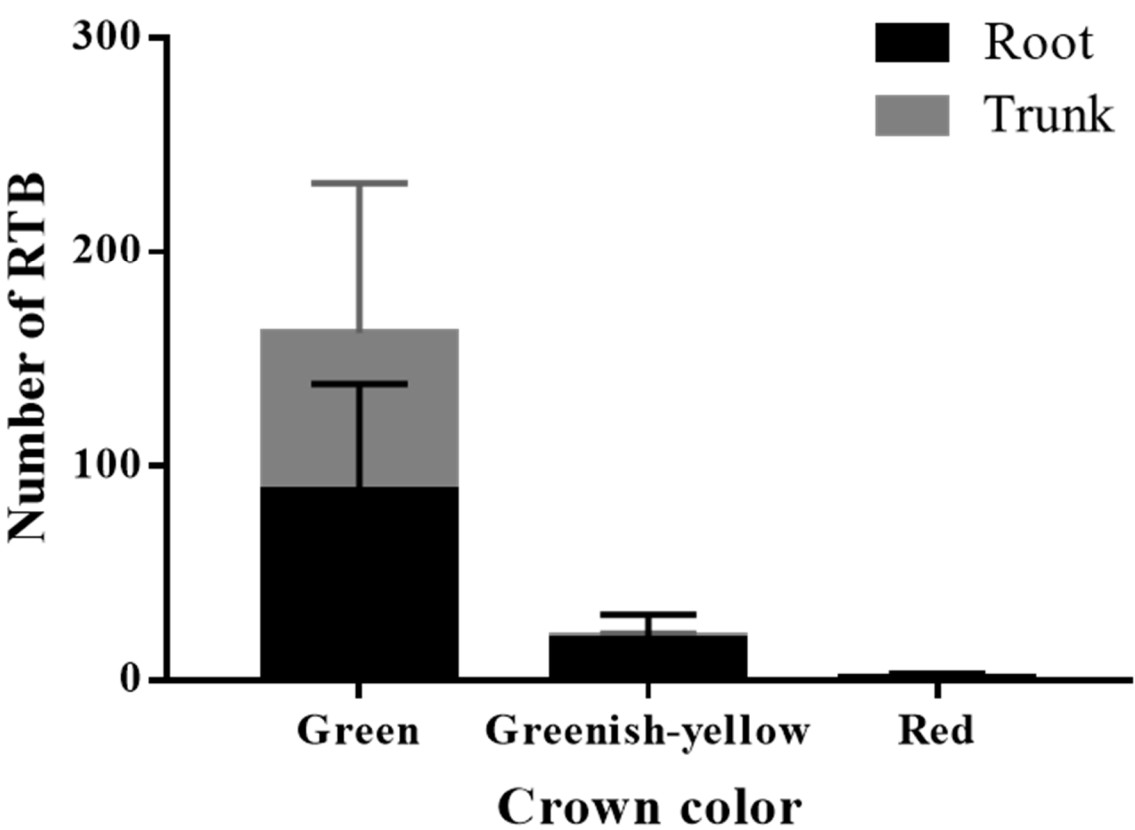

**Figure 3.** Number of RTB in infested trees with different-colored crowns.

**Table 1.** The assessment indicators of pine health status and the number of samples.

| Class | Crown Color | Infestation Symptoms of RTB | Number of Samples |
|---|---|---|---|
| Healthy | Green | No | 200 |
| Infested | Green | Yes | 190 |
| | Greenish-yellow | Yes | 10 |
| Dead | Red | Yes | 138 |
| | Gray | Yes | 62 |

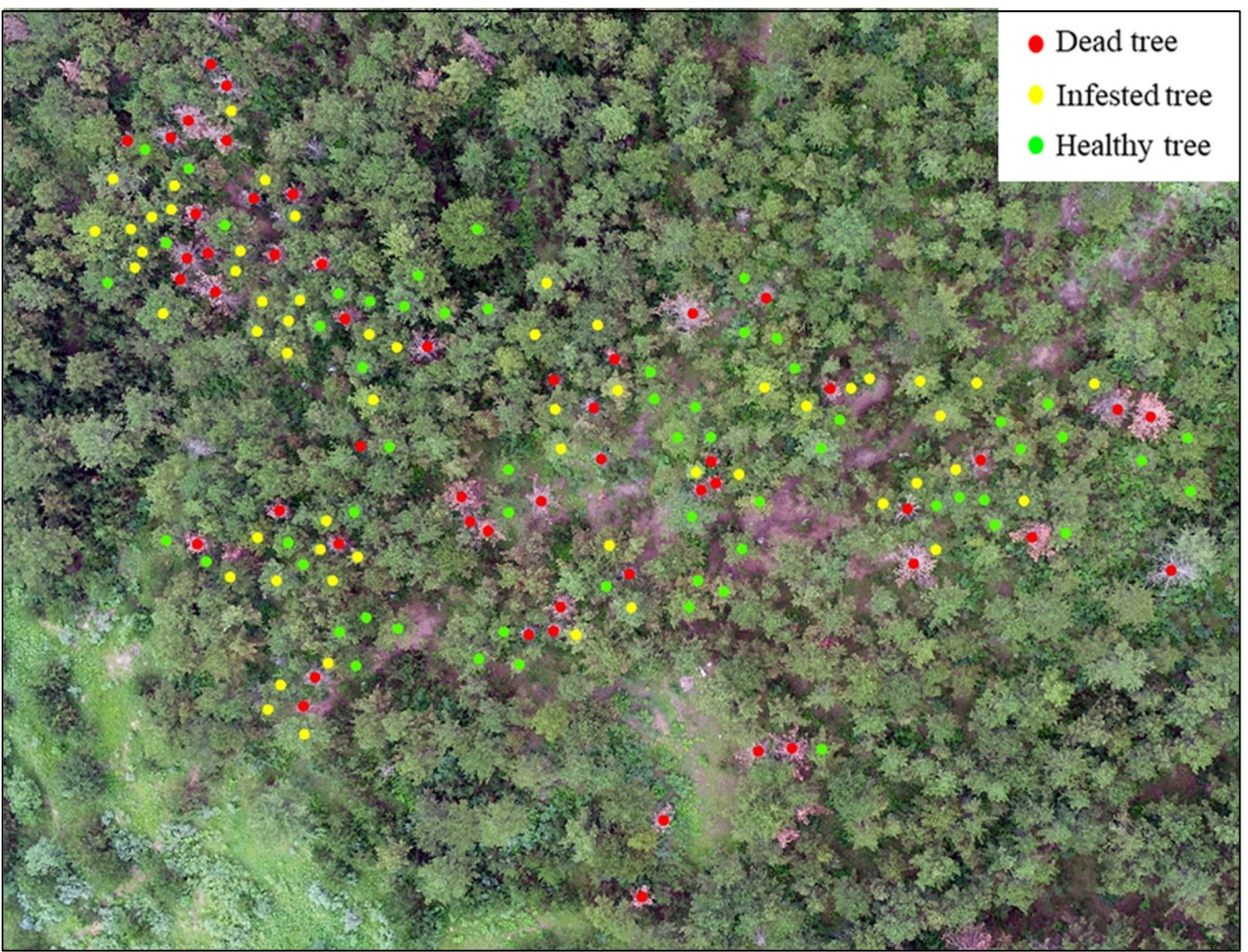

**Figure 4.** A high-resolution orthophoto with dots representing the health status of trees assessed through the field survey. Red dots: dead trees; yellow dots: infested trees; green dots: healthy trees.

*2.3. Features Extraction and Analysis*

To prepare the images and labels for the classification models, we visually interpreted and manually delineated individual tree canopies using the rectangular boxes of the Region of Interest (ROI) Tool in ENVI 5.3. The orthophotos with ground truth data in Section 2.2 were used as a reference for labeling the HSI. A total of 600 trees (200 per class, Table 1) were selected and their distribution is shown in Figure 5. The data cube of each individual tree was cut from HSI using each rectangular box in python with the 'arcpy' package.

The average reflectance of each tree was extracted and then used to calculate the derivatives and SVIs, which were considered to reduce the influence of geometry and light conditions and enhance the ability to detect stress [23,51]. The first and second derivatives of the spectral curve were calculated in OriginPro 2019b (OriginLab Corporation, Northampton, MA, USA). Sixteen SVIs (Table 2) that were found to be important for distinguishing RTB-infested pines at the needle level were calculated [16]. The spectral features of different health categories were compared with each other using the Kruskal–Wallis (K-W) one-way analysis of variance with a *p*-value of 0.01 in R v4.0.4 [21,52].

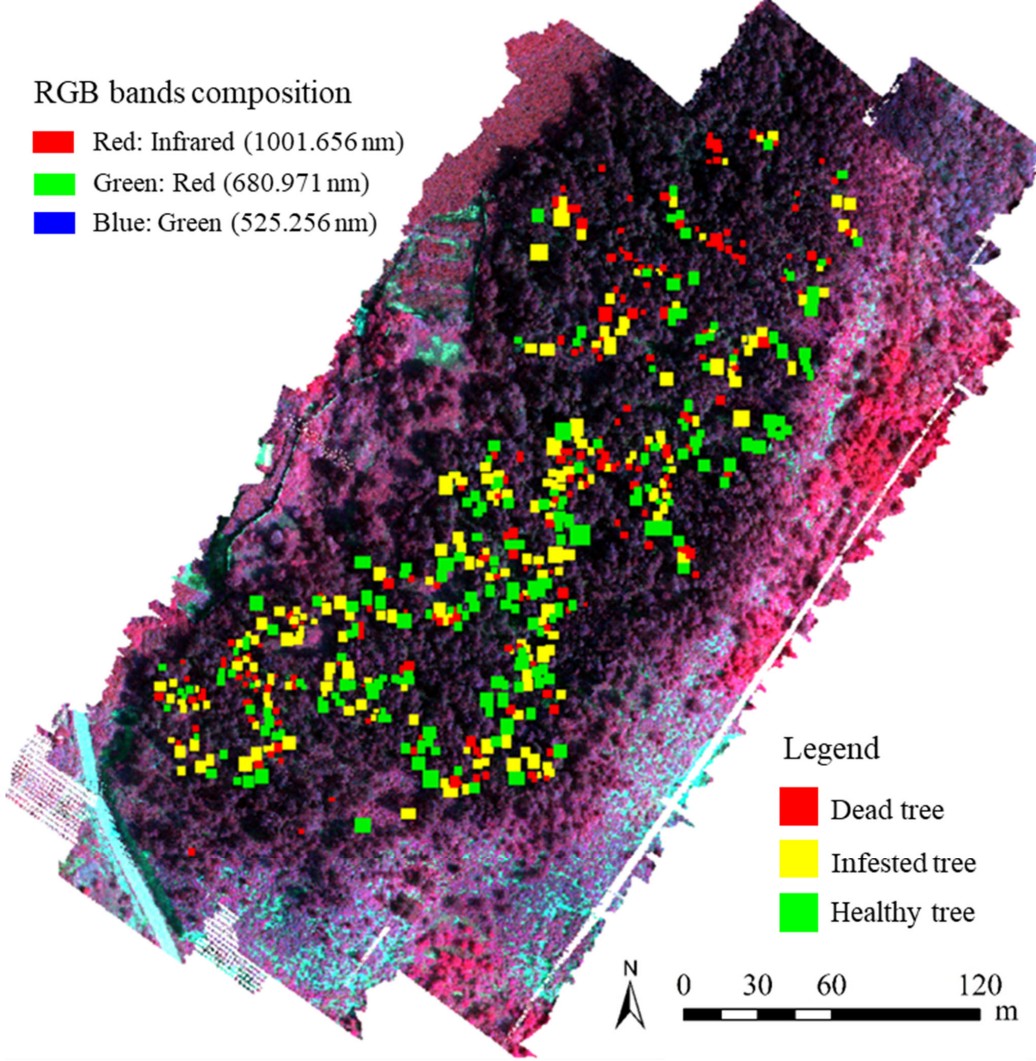

**Figure 5.** Standard false-color image of the stand and distribution of sample trees.

**Table 2.** Spectral vegetation indices (SVIs) used in the study.

| Index | Formulation or Depiction | Bands | Ref. |
|---|---|---|---|
| Simple ratio indices | $SR_1 = R_{546}/R_{538}$ | 40, 38 | [53] |
| Pigment-specific simple ratio | $PSSRc = R_{802}/R_{472}$ | 100, 22 | [54] |
| Ratio analysis of reflectance spectra | $RARS = R_{745}/R_{513}$ | 87, 32 | [55] |
| Lichtenthaler index | $LIC = R_{439}/R_{741}$ | 14, 86 | [56] |
| $D_{737}$ | First derivative of reflectance spectrum at 737 nm | 85 | [57] |
| A ratio of first derivative values | $Voge = D_{715}/D_{707}$ | 80, 78 | [58] |
| First derivative difference index | $DID = D_{1024} - D_{877}$ | 150, 117 | [53] |
| Physiological reflectance index | $PRI_{517} = (R_{517} - R_{534})/(R_{517} + R_{534})$ | 33, 37 | [59] |
| Photochemical reflectance index | $PRIm_2 = (R_{600} - R_{534})/(R_{600} + R_{534})$ | 53, 37 | [60] |

**Table 2.** *Cont.*

| Index | Formulation or Depiction | Bands | Ref. |
|---|---|---|---|
| Red-edge vegetation stress index | $RVSI = (R_{715} - R_{754})/2 - R_{732}$ | 80, 89, 84 | [61] |
| $R_{NIR} \bullet CRI_{550}$ | $R_{771} \times (1/R_{509} - 1/R_{550})$ | 93, 31, 41 | [62] |
| Curvature index | $CUR = R_{677} \times R_{690}/R_{685}{}^2$ | 71, 74, 73 | [63] |
| Health index | $HI = (R_{534} - R_{698})/(R_{534} + R_{698}) - R_{702}/2$ | 37, 76, 77 | [60] |
| Optimal vegetation index | $VI_{opt} = 1.45 \times (R_{802}{}^2 + 1)/(R_{668} + 0.45)$ | 100, 69 | [64] |
| Three-band spectral index | $TBSI = (R_{605} - R_{521} - R_{681})/(R_{605} + R_{521} + R_{681})$ | 54, 34, 72 | [53] |
| Optimized soil-adjusted vegetation index | $OSAVI_2 = (1 + 0.16) \times [(R_{750} - R_{707})/(R_{750} + R_{707} + 0.16)]$ | 88, 78 | [65] |

## 2.4. Classification Models

### 2.4.1. Random Forest (RF)

RF is a robust algorithm based on ensemble learning and estimates by counting the votes of a large number of decision trees, and is widely used in the analysis of remote sensing and hyperspectral data [16,21,23,66–69]. It is considered suitable for solving multi-collinearity problems with a small sample size and large number of variables, and requires fewer user-defined parameters than other algorithms (e.g., SVM) [16,67]. In the study, RF algorithm was used to build classification models to compare the ability of different types of spectral features in detecting an infestation of RTB. The RF classification models were built in the RandomForestClassifier function in Python v3.9.12 with the package 'sklearn'; two important parameters were optimized by a grid search and determined according to the results of a ten-fold cross-validation, which were the number of decision trees in the forest (n_estimators) and the number of features to consider when looking for the best split (max_features) [70]. The n_estimators parameter was tested from values of 100, 500, 1000, 1500 and 2000, and max_features was selected from the values of 'auto', 0.2, 0.4, 0.6, 0.8 and None, in which None and 'auto' represented all variables or the square root of all variables were used at each split, and the remaining values represented the factors multiplied by all variables.

The spectral features with significant differences ($p < 0.01$) in Section 2.3 constituted three datasets (average reflectance, derivatives, SVIs) as input variables for the RF classifier [48]. Three datasets were divided into training and test sets in the ratio 8:2 [39,47,71–73]. The classification models were constructed and tuned on the training sets, and the final models were evaluated unbiasedly on the test sets.

### 2.4.2. Convolutional Neural Network (CNN)

CNN is the most popular deep learning algorithm for remote sensing image analysis, widely used in feature extraction and classification tasks [74]. In this study, a CNN model was developed with reference to the Inception-ResNet-v2 architecture [75] and the overall architecture is shown in Figure 6. Firstly, the cropped hyperspectral cubes of crowns in Section 2.3 were resampled to $32 \times 32 \times 150$ as the input layer of the CNN model using the bilinear interpolation method, where 32 was the average size of the crowns. Then, the input layer was passed through a feature map extraction module to obtain the deep features. Three types of Inception-resnet blocks were introduced into the module. Finally, an average pooling layer and a softmax layer were used to predict the category of bark beetle attacks. A categorical cross-entropy loss between the input and the predicted category was calculated as the loss function for the CNN model and minimized using the Adaptive Moment Estimation (Adam) optimization function through the backpropagation algorithm [39].

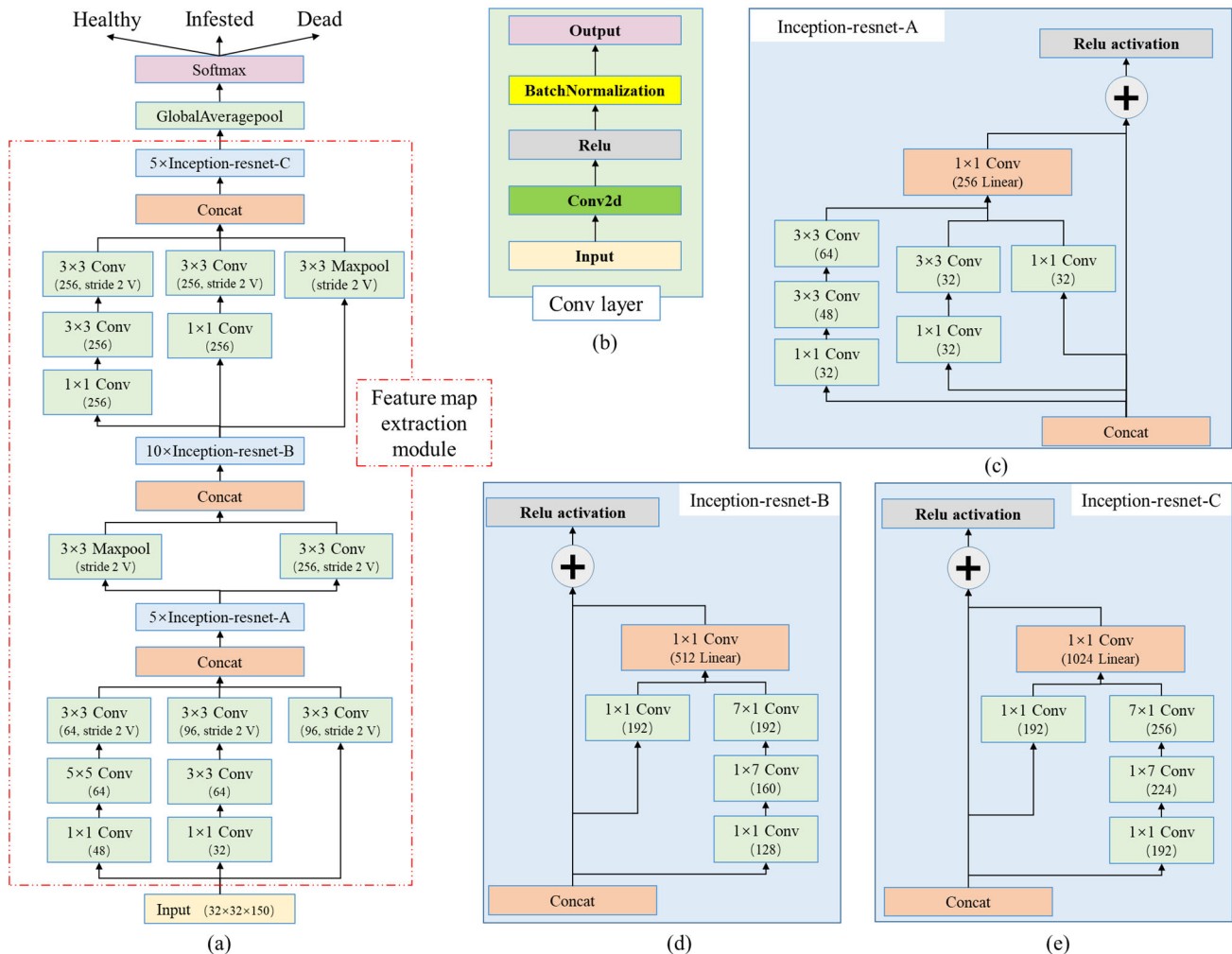

**Figure 6.** Schematics of the CNN model. (**a**) The overall architecture of the model; (**b**) convolution layer architecture containing a 2D convolution, Relu activation function and BatchNormalization layer; (**c**–**e**) Inception-resnet blocks.

Data augmentation is considered an effective strategy to solve the problem of insufficient training samples and restrain overfitting [39,46]. Minařík et al. [48] determined that data augmentation can improve classification accuracy and reduce misclassification. In our study, the dataset consisting of 600 tree crowns was randomly divided into training and test sets with a ratio of 8:2. The samples in the training set were transformed by vertical/horizontal flips or random rotation with 30° intervals [47]. After data augmentation, the training set contained 960 samples, of which 80% were used to train the model and the remaining 20% for internal validation. The CNN model was trained using the Adam optimizer with a batch size of 64. The learning rate was initially set to $10^{-3}$ and finally decreased to $10^{-6}$ with the increase in training iterations. The number of training epochs was set to 100 and early stopping was used to avoid overfitting during the training process. The test set was independent and used to externally evaluate the performance and generalization ability of the final model.

All code was written in Python v3.9.12 with the packages 'numpy', 'osgeo', 'scipy', 'sklearn', 'keras' and 'tensorflow'. The training procedure was implemented based on Tensorflow framework and performed on an NVIDIA GeForce RTX 2080Ti GPU with 11 GB RAM.

### 2.4.3. Evaluation of Models Performance

The confusion matrices were calculated by comparing the predicted classes of the test set with the ground truth classes to evaluate the performance of the classification models. Several metrics could be obtained from the confusion matrix. Overall accuracy (OA) and Cohen's kappa coefficient were used to assess the general performance of the classification models [49,76]. For each infestation class, recall (also called sensitivity), precision and F1-score were calculated based on true positives (*TP*), false positives (*FP*), true negatives (*TN*) and false negatives (*FN*) [39]. The formulas of the metrics are as follows:

$$\text{Recall} = \frac{TP}{TP + FN} \tag{1}$$

$$\text{Precision} = \frac{TP}{TP + FP} \tag{2}$$

$$\text{F1-score} = \frac{2TP}{2TP + FP + FN} \tag{3}$$

$$\text{OA} = \frac{TP + TN}{TP + TN + FP + FN} \tag{4}$$

$$\text{Kappa} = \frac{OA - eAccuracy}{1 - eAccuracy} \tag{5}$$

$$eAccuracy = \frac{\sum_{i=1}^{k} N_P \times N_t}{S^2} \tag{6}$$

where $k$ is the number of categories, $N_p$ is the number of predictions, $N_t$ is the number of truth and $S$ is the number of samples.

## 3. Results

### 3.1. Differences in Spectral Features of Three Health Classes

The changes in tree spectra after being infested by RTB are shown in Figure 7. The reflectance curves of dead trees were significantly different from the other two categories both in the visible and near-infrared (NIR) regions (Figure 7a). The features related to pigment absorption on the spectral curve almost disappeared, and the reflectance decreased significantly in the NIR region. However, the spectral reflectance curve of infested trees displayed a similar overall trend to that of healthy trees. Statistical analysis showed that the spectral reflectance of infested and healthy trees was significantly different in 67 bands in the visible region, and the reflectance of infested trees was higher than that of healthy trees in the 'green peak' (520–588 nm) and red (634–732 nm) regions (Figure 7a). In addition, the position of the 'green peak' of infested trees shifted towards a longer wavelength (from 550 nm to 554 nm).

The derivatives reflected the shape of spectral curves (Figure 7b,c). Compared with healthy trees, the slope (first derivative) and concavity (second derivative) of the spectral curve of dead trees changed significantly in both the visible and NIR regions, while the changes of infested trees were concentrated in the visible region and the amplitude was small. The K-W test (*p < 0.01*) results showed that 55 first derivatives and 52 second derivatives were significantly different between any two classes, among which 4 first derivatives and 2 second derivatives were located in the NIR region. The slope of infested trees increased in the red edge region (680–706 nm), but its red edge position (i.e., the wavelength corresponding to the maximum value of the first derivative in 680–760 nm) did not change significantly.

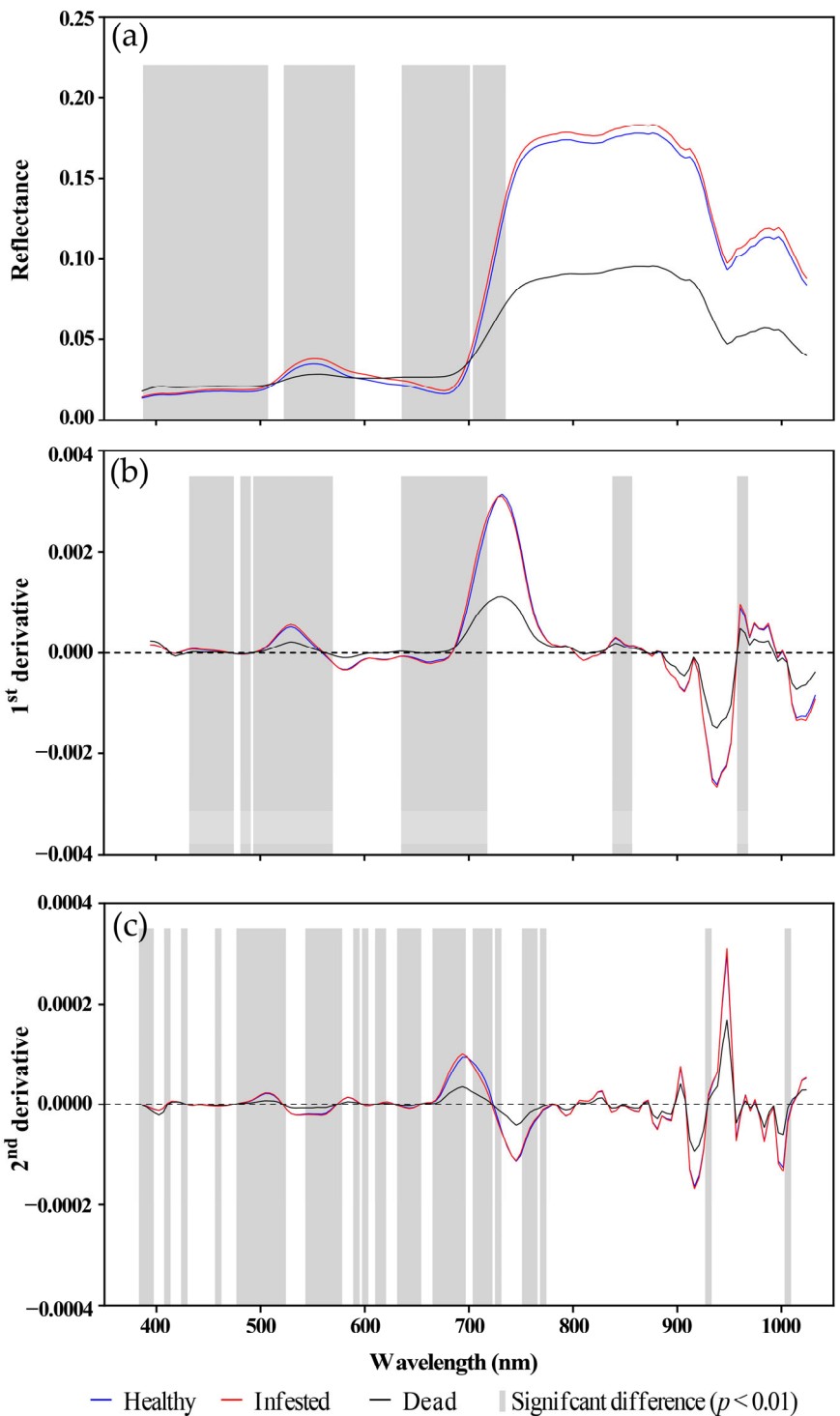

**Figure 7.** Mean crown spectra with 1st and 2nd derivatives. The blue, red and black lines indicate healthy, infested and dead trees, respectively, and grey areas indicate significant differences between any two groups. (**a**) Average spectral reflectance; (**b**) average first derivatives; (**c**) average second derivatives.

The separability between the three health states for individual SVIs is shown in Figure 8. The K-W analysis showed that all indices of the dead trees were significantly different from those of other classes (*p < 0.01*). Almost all indices were segregated at the 25th–75th percentiles except for 'Voge'. Compared with healthy trees, eight SVIs (PSSRc, RARS, Voge, DID, CUR, HI, $VI_{opt}$ and $OSAVI_2$) of infested trees decreased significantly, and

$SR_1$, $PRIm_2$ and TBSI increased significantly. However, these indices partially intersected in the 25th and 75th percentiles between the infested and healthy trees, suggesting that it was difficult to identify early RTB infestation by thresholds based on individual indices. In general, with the severity of damage, seven SVIs (PSSRc, RARS, Voge, CUR, HI, $VI_{opt}$ and $OSAVI_2$) decreased steadily, while PRIm2 increased steadily, and the remaining three SVIs ($SR_1$, DID and TBSI) showed opposite performance in the early and late stages.

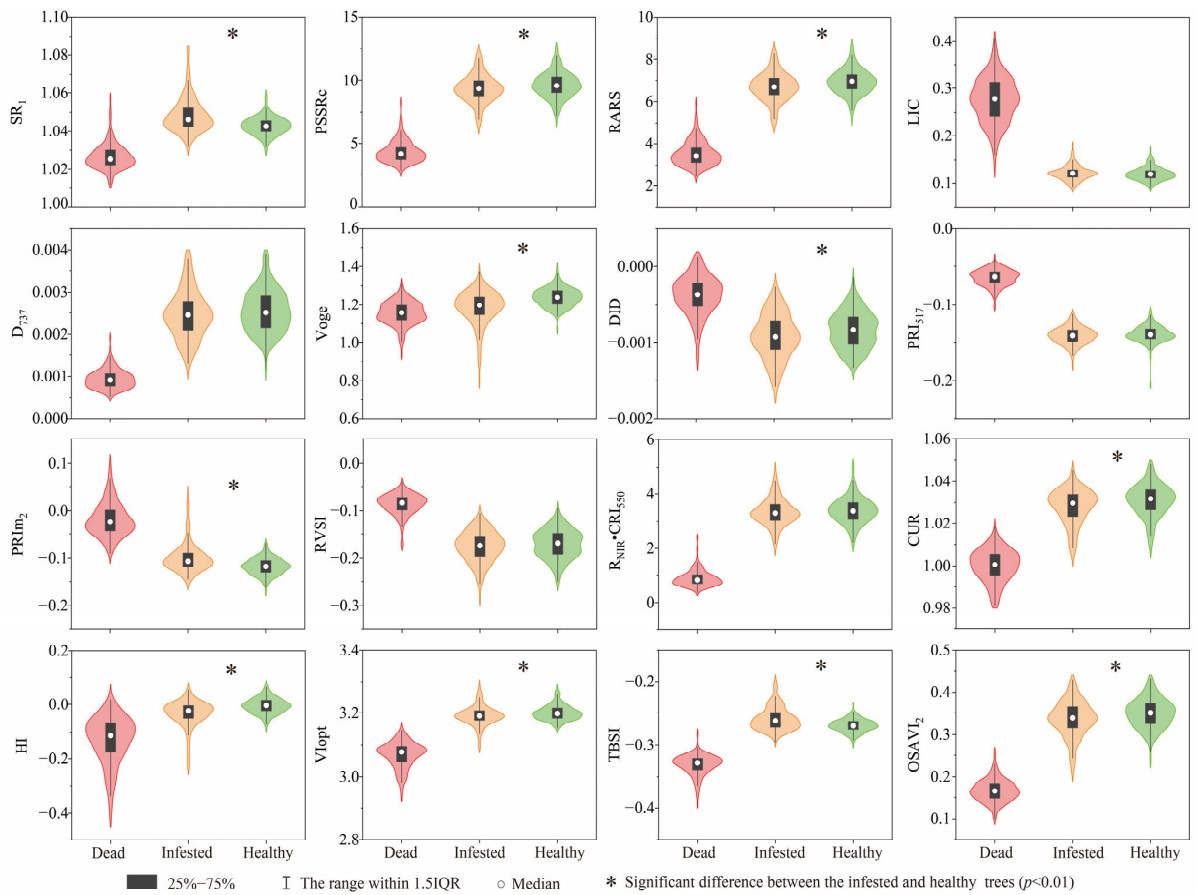

**Figure 8.** Spectral vegetation indices of three health classes. The symbol * indicates statistically significant differences between healthy and infested trees.

*3.2. Classification Results*

Three types of variables (reflectance, derivatives and SVIs) with significant differences ($p < 0.01$) in Section 3.1 were input into the RF classifier to establish classification models. The number of input variables and the parameters optimized by 10-fold cross-validation (10-k CV) were shown in Table 3. The validation loss of the CNN model stabilized and stopped training in 84 epochs, with an internal validation accuracy of 85.42%.

**Table 3.** Parameters of the RF models with different input variables.

| Model | Input Variables | Number of Variables | Parameters | |
|---|---|---|---|---|
| | | | n_Estimators | Max_Features |
| RF_R | Reflectance of bands | 67 | 1000 | 0.4 |
| RF_D | 1st and 2nd derivatives | 107 | 2000 | 0.4 |
| RF_S | SVIs | 11 | 1000 | None |

Figure 9 showed the classification results of three RF models and a CNN model on the test sets. The overall results of the CNN model were the best, with an OA of 83.33% and a kappa of 0.75. For the RF classifiers, the model using SVI as input variables (RF_S) performed second best with an OA of 82.5% and kappa of 0.6, while models using the other two input datasets (RF_R and RF_D) performed poorly with an OA of less than 80%; the main errors were mismatches between the infested and healthy trees.

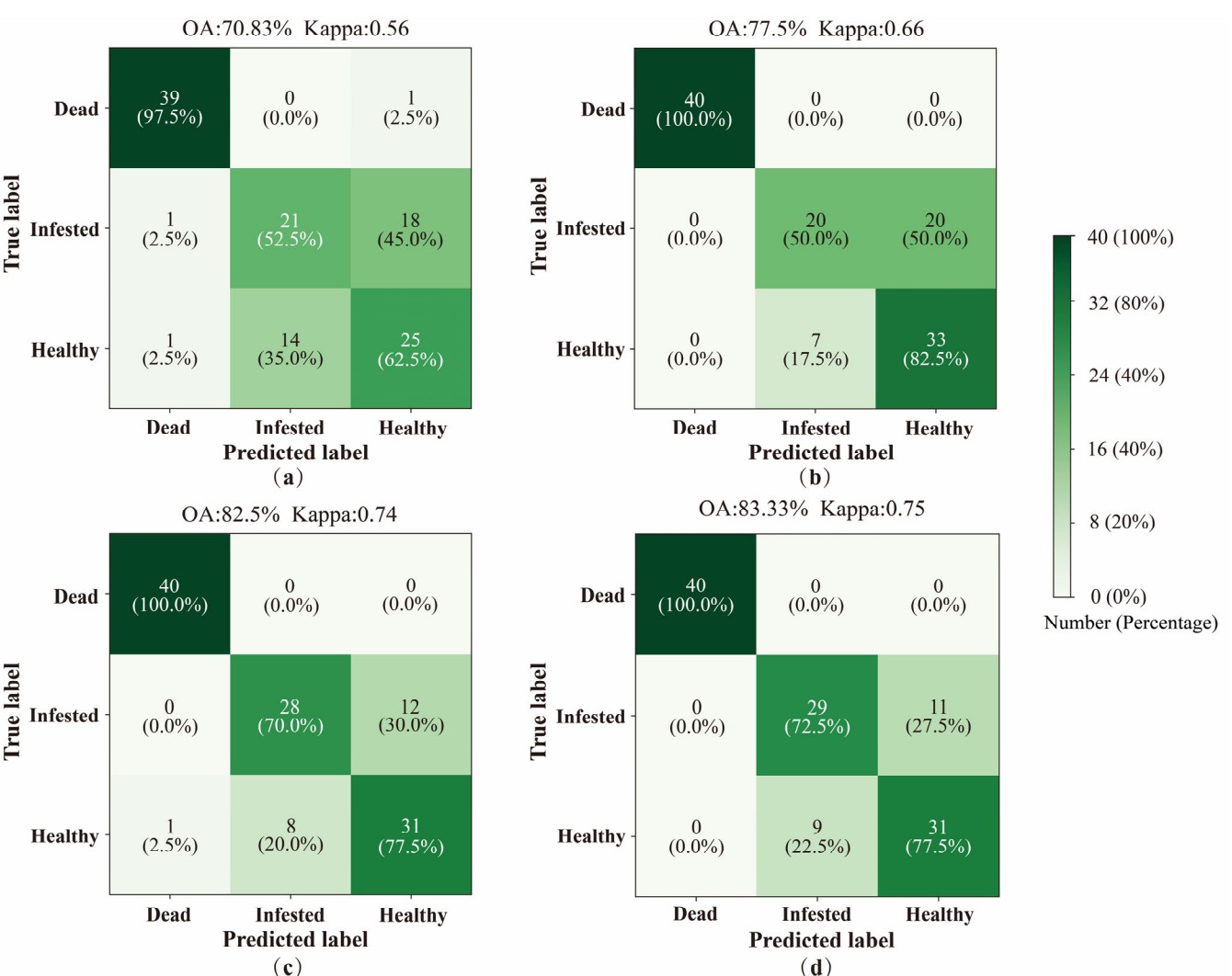

**Figure 9.** General performance and confusion matrices of the RF and CNN classification models. (**a**) RF_R: RF model with reflectance values as input variables; (**b**) RF_D: RF model with derivatives as input variables; (**c**) RF_S: RF model with SVIs as input variables; (**d**) CNN model.

The ability of different models to identify each health category is shown in Figure 9 and Table 4. For dead trees, all models almost accurately detected them, and all evaluation metrics were greater than 95%, especially in the RF_D model and CNN model, whose recall and precision both reached 100%, meaning that there were neither commission nor omission errors. For infested trees, the CNN and RF_S models performed better than the other two models, with F1-scores higher than 70%. The recall values for the RF_R and RF_D models were 52.5% and 50%, respectively, meaning that about half of the infested trees were missed. These missed infested trees will become the source of the next year's infestation. For healthy trees, the RF_R model performed the worst, with an F1-score of 59.52%, recall of 62.5% and precision of 56.82%, representing that 37.5% of healthy trees were misdiagnosed as trees attacked by RTB, and 43.18% of trees attacked by RTB were considered healthy

trees. The RF_D model's recall was the highest, reaching 82.5%, but its precision was only 62.26% due to the misclassification of half of the infested trees (Figure 9b). The RF_S and CNN models performed similarly in the classification of healthy trees, with all three metrics greater than 70%.

**Table 4.** Evaluation metrics for each infestation class.

| Model | Input | Class | Recall (%) | Precision (%) | F1-Score (%) |
|-------|-------|-------|-----------|---------------|--------------|
| RF_R | Reflectance values | Healthy | 62.5 | 56.82 | 59.52 |
| | | Infested | 52.5 | 60 | 56 |
| | | Dead | 97.5 | 95.12 | 96.3 |
| RF_D | 1st and 2nd derivatives | Healthy | 82.5 | 62.26 | 70.97 |
| | | Infested | 50 | 74.07 | 59.7 |
| | | Dead | 100 | 100 | 100 |
| RF_S | SVIs | Healthy | 77.5 | 72.09 | 74.7 |
| | | Infested | 70 | 77.78 | 73.68 |
| | | Dead | 100 | 97.56 | 98.77 |
| CNN | HSIs | Healthy | 77.5 | 73.81 | 75.61 |
| | | Infested | 72.5 | 76.32 | 74.36 |
| | | Dead | 100 | 100 | 100 |

## 4. Discussion

The objective of this study was to evaluate the use of high-resolution HSI for detecting RTB infestations at the individual tree level. The results showed that the spectral features of pine trees change significantly after an RTB infestation, and they can be detected at the early stage using appropriate spectral characteristics and algorithms.

Unsurprisingly, the spectral characteristics of dead trees change very noticeably (Figure 7) due to the fading and shedding of needles. Changes in the visible region are mainly due to chlorophyll degradation, while significant declines in the NIR region are associated with changes in canopy parameters and leaf structure [52,77]. There is no doubt that dead trees (in the red and gray attack stages) are easily detected based on spectral signatures. Our classification results showed that even if only the visible band reflectance values were used as the input variables (RF_R model), the detection accuracy of dead trees could reach more than 95% (Table 4).

Detecting trees at the early attack stage is more challenging. The spectral reflectance and derivative curves of infested trees show similar trends to those of healthy trees (Figure 7). The K-W analysis showed that the reflectance of infested trees significantly increased in the 'green peak' and red regions, and the position of the 'green peak' shifted towards longer wavelengths. These changes have also been found in [21,23,78] and are thought to be related to changes in pigment content. However, there was no significant difference in the NIR region between the infested and healthy trees, which was inconsistent with our previous results at the needle level [16], in which we found that the spectral reflectance of infested trees decreased evidently in the NIR plateau region. The canopy spectrum is a synthesis of all individual leaves and background surface spectra including soil, understory, branches, leaf-layering and shadows within crown [79]. Such inconsistent results may be attributed to the canopy contribution to the NIR reflectance masking the influence of internal structural changes in the needles. Furthermore, in order for the CNN model to take advantage of spatial geometric information, the crowns we sketched were square without excluding background and shadow pixels, which have a more severe effect on NIR than red [79]. The time difference in acquiring the spectrum may be another reason. Bárta et al. [78] monitored a bark beetle infestation using time series of HSIs and found that the wavelengths associated with chlorophyll absorption were affected first, followed by the NIR region.

Due to the similarity of the spectral curves between the infested and healthy trees, the RF models using reflectance and derivatives as input data cannot effectively distinguish between the two groups (Table 4); the overall accuracy of the RF_D model was higher than

that of the RF_R model, but it had the lowest recall for infested trees and an overestimation for healthy trees. Derivative analysis focuses on the shape of spectral curves [51], so the ability to identify infested trees was poorer, but dead trees could be accurately recognized (Figure 9b). SVIs were considered to be better stress indicators and less affected by illumination and background [79], as supported by the classification results of the RF_S model using SVIs as input data. Näsi et al. [80] also found that spectral indices provided better results than the full spectrum using a k-nearest-neighbor classifier. Some of the SVIs used in this study might also be used for the early detection of RTB damage on a large scale based on spectral trajectories of satellite data [81]. For example, TBSI, PSSRc, RARS, $OSAVI_2$, $PRIm_2$ and $VI_{opt}$ could be calculated using the corresponding bands of Sentinel-2. Our proposed CNN model achieved classification accuracy comparable to the RF_S model, with an overall accuracy of 83.33% and a recall rate of 72.5% for the early infestation stage, which was comparable to the accuracy of previous studies on the detection of other forest stress [20,21,80,82,83]. However, the high dimensionality and information redundancy of hyperspectral data led to the complexity of the model and large computational amount (16,473,027 trainable parameters), which limited the capability of real-time detection. Our results showed that visible bands had significant differences between healthy and early infested trees (Figure 7). Therefore, the use of cheaper RGB-Red Edge multispectral cameras combined with pre-trained CNNs can be investigated in the future to early detect RTB infestation in real time.

The time it takes for trees to start discoloring after an infestation is variable and is affected by many factors, such as the number of beetles, the genetics and vitality of the tree and the environmental conditions [50,78]. A later detection time might improve accuracy, for example, in the early spring of the following year [16,23], but it would also mean a shorter time to take pest control action, as RTBs reach their flight peak in mid-May. Therefore, a study on the proper timing of data acquisition, taking into account detection accuracy, pest biology, tree response time and logistics factors, needs to be conducted.

Deep learning algorithms can learn "end-to-end", automatically extracting features from training data without the need for complex feature engineering and prior expert knowledge, such as calculation and screening of vegetation indices. Previous studies have shown that CNN algorithms outperform traditional machine learning algorithms in tree species classification and plant pest detection [39,42,43,45,48]. In this study, our proposed CNN model was also the best performer. Some studies have proposed that 3D-CNN-based models are more suitable for hyperspectral image classification and have better accuracy than 2D-CNN-based models [29–32,34,49]. However, these models may not be suitable for UAV hyperspectral images with high spatial resolution, because if model inputs are small neighboring regions, some spatial information would be lost, and if the model inputs are individual trees, the time cost and hardware requirements for training would be too high. In this study, we manually segmented the individual crowns using rectangular boxes before the classification due to the limited number of samples. Numerous studies have developed individual crown segmentation algorithms based on point cloud data or canopy height models (CHM) [41,47,48]. Miyoshi et al. [44] proposed a deep learning method to identify individual tree species using UAV-based HSIs. Instance segmentation is developing rapidly in the field of computer vision, and some algorithms have been introduced into forestry, such as Mask R-CNN for tree detection [84]. In the future, with adequate sample collection, segmentation and classification can be combined to develop a model that can automatically detect infested trees at an early stage.

## 5. Conclusions

This study tested the potential of UAV high-spatial-resolution hyperspectral imagery to detect RTB-infested pine trees. We compared the performance of different spectral features and models in classifying pines into three health states: healthy, infested (at the green or yellow attack stage) and dead (at the red or gray stage) trees. The main conclusions are as follows:

(1) The spectra of pine crowns markedly changed after a RTB infestation. Compared to healthy trees, the spectral curves of dead trees changed significantly in both the visible and NIR regions, while the difference of infested trees was significant only in the visible region. All 16 SVIs used in this study were significantly different for dead trees, whereas 11 were significantly different for infested trees.

(2) The model using SVIs as variables performed better than the other two models when the reflectance, first and second derivatives, and SVIs were input into the random forest (RF) classifier.

(3) The CNN model performed best in classifying bark beetle disturbances, with an overall accuracy of 83.33% and a recall rate of 72.5% for early infested trees.

Our results demonstrated that a UAV-based hyperspectral image can be successfully used for the detection of a RTB infestation at the individual tree level. The SVIs and classification models used in the study can provide a reference for the early detection of bark beetle damage. CNN is suitable for the detection of bark-beetle-infested trees from hyperspectral images. Future research could develop a CNN-based model for the automatic early stage identification of individual trees infested by bark beetles by combining segmentation and classification tasks.

**Author Contributions:** Conceptualization, Y.L. and L.R.; methodology, B.G. and L.Y.; software, B.G.; validation, B.G.; formal analysis, B.G.; investigation, B.G. and Z.Z.; writing—original draft preparation, B.G.; writing—review and editing, B.G., L.Y. and L.R.; project administration, Y.L.; funding acquisition, Y.L. All authors have read and agreed to the published version of the manuscript.

**Funding:** This research was funded by National Key R & D Program of China (2022YFD1400400) and National Key R & D Program of China (2022YFD1401000).

**Institutional Review Board Statement:** Not applicable.

**Informed Consent Statement:** Not applicable.

**Data Availability Statement:** Not applicable.

**Acknowledgments:** The authors would like to thank Lingyuan City Forest Pest Control and Quarantine Station and Ningcheng County Heilihe Forest Farm for their support in the field investigation.

**Conflicts of Interest:** The authors declare no conflict of interest.

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
