# Peer review of "Early Detection of Dendroctonus valens Infestation at Tree Level with a Hyperspectral UAV Image"

_remotesensing, doi:10.3390/rs15020407_

Round 1
Reviewer 1 Report
1) Some abbreviations have no explanation, eg UAV, RTB (in abstract)
2) How is the real-time analysis of the proposed study?
3) Why compare CNN with RF? Besides RF, SVM can also be used.
4) Contributions of the study should be specified in the introduction section as items.
5) CNN-based studies on the detection of agricultural pests should be cited.
6) Figure 8 should be explained in more detail.
7) The formulas of the metrics in Table 4 should be added to the article.
Author Response
Dear reviewer,
We really appreciate your comments on our paper. We have revised the manuscript according to your kind advices and responded to the comments point-by-point.
Please see the attachment.

Reviewer 2 Report
Dear Authors,
Your manuscript entitled “Early detection of Dendroctonus valens infestation at tree level with a hyperspectral UAV image” describes the application of UAV-acquired hyperspectral data to detect early pest infestations red turpentine beetle on trees. The spectral characteristics of Pinus tabuliformis in three states (healthy, infested and dead) was compared and established classification models using three groups of features (reflectance, derivatives and spectral vegetation indices) and two algorithms (random forest and convolutional neural network).
The article is very interesting, although several things should be improved.
Line 73-74 - what was the spatial resolution of the data and the spectral range
In parts 58-87 there should be details about the type of hyperspectral data, pixel size and spectral range
Line 184 - why was this 8:2 data split used? have others been tested?
Figure 6 - improve quality
Figure 7 - better mark statistically significant ranges with a shade of grey on the whole figure
Analysis of derived results is missing.
Figure 9 - larger font size. Kappa values to 2 decimal places
Line 338-339 - please specify which ones, instead of "some of the SVIs"
Sincerely
Reviewer
Author Response

(The authors gave the same response as above.)

Reviewer 3 Report
The authors present a novel research to provide early detection of red turpentine beetle infestation (TBI) on Chinese pine trees using machine learning methods such as Random Forests and Convolutional Neural Networks (CNNs) and high-resolution hyperspectral imagery taken using UAVs.
The methods employed are adequate, where cross-validation was used to find the best model parameters and a separate test set was used to test the machine learning algorithms. Spectral bands, derivatives of the spectral bands and vegetation indexes were adequately explored in the research. The methods used to test the hypothesis are adequated.
The Figures and Tables are adequate, however when introducing Table 3, the Number of Variables used should be properly explained, since there are 150 spectral bands and the number of variables is lower (higher significance), also explain the number of SVIs used in this table.
The conclusion indicates that only the visible band shows significant differences between healthy and infested (but not dead) trees. Hence, there should be a discussion of this, since the use of much cheaper RGB cameras or even RGB-Red Edge Multispectral cameras could be used in the future to perform early detection of TBI on pine trees using also CNNs pre-trained with RGB images.
The results are reproducible, although hyperspectral cameras on UAVs are too expensive to be used routinely further research should be done with RGBs or Multispectral cameras.
Author Response

(The authors gave the same response as above.)

Reviewer 4 Report
The authors proposed a really interesting topic, utilizing UAV hyperspectral data to map Dendroctonus valens with the title of “Early detection of Dendroctonus valens infestation at tree level 2 with a hyperspectral UAV image”. The research is well conducted and documented, so that I recommend minor revision. Several minor comments to this research are as follows:
1. The title suggesting that you are able to identify early infested trees, while in fact the classes are dead, infested and healthy. I suggest the change the title to better represent the content of the research;
2. Lack of literature studies regarding the hyperspectral classification such as spectral angle mapper, spectral mixture analysis and multiple endmember SMA. Please add a background or discussion on why such methods are not used in this study, and why do you prefer to use random forests instead.
3. On the accuracy assessment, it is true that the accuracy of CNN is higher than the others. But on the validation data, the RF_D was higher. Can you please explain why do you think CNN is better than the other classifier when CNN shows an overfitting result?
Author Response

(The authors gave the same response as above.)

Round 2
Reviewer 1 Report
The authors answered the comments with precision. This article is acceptable.